# Assessing the Utility of ColabFold and AlphaMissense in Determining Missense Variant Pathogenicity for Congenital Myasthenic Syndromes

**DOI:** 10.3390/biomedicines12112549

**Published:** 2024-11-08

**Authors:** Finlay Ryan-Phillips, Leighann Henehan, Sithara Ramdas, Jacqueline Palace, David Beeson, Yin Yao Dong

**Affiliations:** 1Nuffield Department of Clinical Neurosciences, University of Oxford, Oxford OX3 9DS, UK; 2Neurology Department, John Radcliffe Hospital, Oxford OX3 9DU, UK; 3Department of Paediatric Neurology, John Radcliffe Hospital, Oxford OX3 9DU, UK; sithara.ramdas@ouh.nhs.uk; 4MDUK Neuromuscular Centre, Department of Paediatrics, University of Oxford, Oxford OX3 9DU, UK

**Keywords:** CMS, AlphaFold2, ColabFold, AlphaMissense, pathogenicity, prediction

## Abstract

Background/Objectives: Congenital myasthenic syndromes (CMSs) are caused by variants in >30 genes with increasing numbers of variants of unknown significance (VUS) discovered by next-generation sequencing. Establishing VUS pathogenicity requires in vitro studies that slow diagnosis and treatment initiation. The recently developed protein structure prediction software AlphaFold2/ColabFold has revolutionized structural biology; such predictions have also been leveraged in AlphaMissense, which predicts ClinVar variant pathogenicity with 90% accuracy. Few reports, however, have tested these tools on rigorously characterized clinical data. We therefore assessed ColabFold and AlphaMissense as diagnostic aids for CMSs, using variants of the CHRN genes that encode the nicotinic acetylcholine receptor (nAChR). Methods: Utilizing a dataset of 61 clinically validated CHRN variants, (1) we evaluated the possibility of a ColabFold metric (either predicted structural disruption, prediction confidence, or prediction quality) that distinguishes variant pathogenicity; (2) we assessed AlphaMissense’s ability to differentiate variant pathogenicity; and (3) we compared AlphaMissense to the existing pathogenicity prediction programs AlamutVP and EVE. Results: Analyzing the variant effects on ColabFold CHRN structure prediction, prediction confidence, and prediction quality did not yield any reliable pathogenicity indicative metric. However, AlphaMissense predicted variant pathogenicity with 63.93% accuracy in our dataset—a much greater proportion than AlamutVP (27.87%) and EVE (28.33%). Conclusions: Emerging in silico tools can revolutionize genetic disease diagnosis—however, improvement, refinement, and clinical validation are imperative prior to practical acquisition.

## 1. Introduction

Congenital myasthenic syndromes (CMS) are a group of rare mendelian monogenic disorders, characteried by abnormal signal transmission from nerve to muscle, across the neuromuscular junction (NMJ) [1]. Fatigable weakness of the skeletal muscles is the main symptom of CMS patients, although severity ranges from mild fatigability to severe disability, respiratory crises, and premature death [2]. Pathogenic variants in >30 causative genes have been identified [3], reflecting the number of proteins essential for the development, maintenance, and function of the NMJ.

Amongst the causative CMS genes identified, variants in the CHRN genes that encode the subunits of the muscle nicotinic acetylcholine receptor (nAChR) are responsible for ~50% of all CMS cases [4]. Muscle nAChR is a pentameric ligand-gated ion channel, comprising α1 (x2), β1, δ, and ε subunits [5]. Following binding of acetylcholine (released from the pre-synaptic nerve terminal), the nAChR opens, permitting an influx of Na^+^ and Ca^2+^ ions into myocytes, initiating an endplate potential that induces muscular contraction [6]. CMS-causing variants of the nAChR do not cause homogenous pathophysiology—rather, different variants disrupt nAChR function and neuromuscular signal transmission in different ways, precipitating three distinct CMS clinical subgroups, classified according to their underlying pathophysiology: receptor deficiency syndromes (RDS—phenotype MIM numbers: CHRNA1 = 253290, CHRNB1 = 616314, CHRND = 616323, CHRNE = 608931), slow-channel syndromes (SCS—phenotype MIM numbers: CHRNA1 = 601462, CHRNB1 = 616313, CHRND = 616321, CHRNE = 605809), and fast-channel syndromes (FCS—phenotype MIM numbers: CHRNA1 = 608930, CHRND = 616322, CHRNE = 616324) [7].

A generic myasthenia diagnosis can be made based upon clinical presentation. However, treatments for certain subtypes have detrimental effects on other subtypes [8]. It is therefore imperative that a genetic diagnosis is made, in which the causative gene, variant, and its pathophysiological effect are elucidated in order to optimize treatment [9]. Rapid advancements in our understanding of human genetics over the last 20 years have made nonsense, frameshift and splice boundary variants easily identified as pathogenic. However, it is still challenging to determine whether missense variants are pathogenic. Whilst many CMS-causing variants are already characterized, novel candidate variants require in vitro expression studies and NMJ functional studies to determine pathogenicity (and if pathogenic, their specific pathophysiological mechanism). This manual process is lengthy, and can only be performed at a handful of clinics worldwide, delaying treatment optimization for patients.

Since 1994, biennial Critical Assessment of Structure Prediction (CASP) competitions have tracked the progress of attempts to predict the 3D shape of proteins from their amino acid sequence [10]. In 2021, DeepMind’s AlphaFold2 recorded unprecedented accuracy in this challenge of structure prediction, demonstrating ‘accuracy competitive with experimental structures’ [11], with further strides made since. Notably, OmegaFold [12] and ESMFold [13] both forego AlphaFold2’s requirement for multiple sequence alignments within the structure prediction process, primarily in order to broaden the scope of applicable proteins in the case of OmegaFold, and increase prediction efficiency with ESMFold. Furthermore, RosettaFold [14] and more recently AlphaFold3 have demonstrated significant advacements in the accurate prediction of protein-ligand interactions [15]. Despite these advances, analysis suggests that AlphaFold2 remains superior with regards to accuracy of single protein structure prediction (not taking into account the most recent AlphaFold3 software) [16].

This novel ability to precisely predict the 3D structure of a protein from its primary sequence has sparked interest in possible applications of AlphaFold2 for predicting variant effects on protein structure, stability, and function. Despite DeepMind claiming that ‘AlphaFold has not been validated for predicting the effect of variants’ [17], with certain studies backing this claim [18,19], others have provided evidence for such a use case [20,21]. ColabFold is an open-source, accelerated version of AlphaFold2, available as a Jupyter notebook within Google Colaboratory that reportedly does not compromise the quality of structure prediction [22]. It does not require large local computing power, nor the extensive bioinformatics skills that Alphafold2 needs to set up, making it a more accessible tool for CMS clinicians worldwide.

In September 2023, DeepMind advanced upon AlphaFold2 to release AlphaMissense, a missense variant pathogenicity prediction model that leverages AlphaFold2’s structure predictions at its core. Upon its release, it was claimed that AlphaMissense predicts variant pathogenicity with 90% accuracy on variants within the ClinVar database [23].

The aim of this study was thus to test whether ColabFold and/or Alphamissense could accurately predict the pathogenicity of missense variants in CHRN genes that produce the different subunits of the muscle nAChR receptor. To do so, we used a clinically and experimentally validated database of genetic variants observed in our UK national congenital myasthenia referral service [24], containing 42 pathogenic variants (including RDS, SCS, and FCS variants), as well as 19 non-pathogenic variants of the nAChR (spanning all adult subunits). For each variant, we ran the wild-type and pathogenic/non-pathogenic variant nAChR subunit sequences through ColabFold, before assessing the variant’s ColabFold ‘signature’; i.e., the variant’s effect on the ColabFold’s prediction from the wild-type subunit to the mutated subunit. This was based on the output data from each ColabFold prediction, which are the same as Alphafold2—the position of amino acid residues, confidence in the predicted position of each amino acid, and the overall predicted quality of template modelling.

We wanted to investigate whether any of these ColabFold data outputs distinguished CHRN CMS variant pathogenicity. Therefore, for each metric, we first tested for differences between pathogenic and non-pathogenic variants, and secondly, we split the pathogenic variants into their respective syndrome groups (RDS/SCS/FCS) and compared them once more.

Subsequently, in order to compare the ability of AlphaMissense to predict CHRN variant pathogenicity with established non-structural in silico methods, we finally assessed the capability of AlphaMissense, AlamutVP, and EVE to differentiate pathogenic from non-pathogenic variants within our dataset. AlamutVP is currently widely used by clinical geneticists, and suggests an ACMG classification [25] for variants based upon a multiplicity of different sources, including computational predictors (REVEL, MaxEntScan, NNSPLICE, SIFT, and PolyPhen-2) and population data [26]. For further comparison, we also tested EVE, an independent, evolutionary-based pathogenicity prediction program, not trained directly on ClinVar, which has recently been demonstrated as a leading program in the field [27].

We hypothesized that our ColabFold signature metrics and/or AlphaMissense may differentiate pathogenic and non-pathogenic CHRN mutations, and thus could be utilized as part of the CMS diagnostic work-up, in order to improve diagnostic waiting times for patients. We find that the Colabfold prediction metrics related to protein prediction confidence do differ significantly between pathogenic CHRN subgroups, but not from the non-pathogenic variants. Furthermore, we show that AlphaMissense does correctly predict CHRN variant pathogenicity more frequently than the existing pathogenicity prediction programs, AlamutVP and EVE. However, it still predicted more than half of the non-pathogenic variants as pathogenic. Our results provide preliminary evidence that new, deep-learning based in silico structural and pathogenicity prediction programs demonstrate potential utility in the diagnosis of genetic disease. That said, considering the high clinical threshold for practical acquisition, we must reject our hypothesis and state that further improvement and refinement of such models is required prior to practical application.

## 2. Materials and Methods

### 2.1. Variant Dataset

We obtained an anonymized dataset from the UK national congenital myasthenia referral service based in Oxford, containing a list of genetic variants that had been tested for CMS pathogenicity, as well as their functional effects and ensuing clinical syndrome (if pathogenic). This dataset was then curated to contain only CHRN variants, of which existed 42 pathogenic (23 RDS, 14 SCS, and 5 FCS) variants, and 19 non-pathogenic variants spanning each nAChR subunit [Appendix A: Variants, Variants Validator].

The National Congenital Myasthenia Service defines variant pathogenicity using functional diagnostic experiments. RDS variants result in <25% wild-type nAChR expression, and SCS variants induce significantly longer opening times than wild-type receptors in single-channel recordings, whilst FCS variants induce significantly shorter opening times than wild-type receptors in single-channel recordings. Non-pathogenic variants were tested by similar expression tests and single-channel recordings, but did not significantly affect receptor expression or kinetics.

### 2.2. ColabFold

Wild-type protein sequences of the nAChR genes were obtained from UniProt (https://www.uniprot.org/), using the canonical sequences of the nAChR subunits (UniProt ID: CHRNA1: P02708-2, CHRNB1: P11230-1, CHRND: Q07001-1, CHRNE: Q04844) (accessed on 4 September 2022). Variant protein sequences were then produced manually by altering wild-type protein sequences appropriately.

We then ran the individual wild-type subunit sequences, and each individual mutated sequence through ColabFold Batch (available at https://colab.research.google.com/github/sokrypton/ColabFold/blob/main/batch/AlphaFold2_batch.ipynb#Instructions) (accessed on 4 September 2022), with parameters outlined in Table 1. Predicted models with the highest global predicted local distance difference test (pLDDT) scores were taken from ColabFold as output for that specific sequence.

### 2.3. Measuring Variant-Induced Predicted Structural Disruption

To calculate the root mean square deviation (RMSD), both wild-type and mutated ColabFold-predicted subunits were loaded into PyMol 2.5.2, before the ‘align’ feature was used, with PyMol providing an RMSD measurement in the process.

Following this, we used the ‘measurement’ tool on PyMol to individually calculate the Euclidean distance between the corresponding residues on the wild-type and mutated predicted nAChR subunit structure. To determine 10aa local distance difference (LDD) scores, we calculated the average distance between the α-carbon of the corresponding amino acids five residues both upstream and downstream of the mutated residue, along with the mutated residue itself. Similarly for 5 Å LDD scores, we measured the average distance between all the corresponding, neighboring residues that fell within a 5 Å 3D radius of the mutated α-carbon atom (as determined by PyMol using the command: ‘select all within 5 of Residue/CA’).

### 2.4. Measuring Variant-Induced Structure Prediction Confidence Changes

Global pLDDT scores were taken from the ColabFold output file and % change was calculated from wild-type to mutated nAChR subunit structure predictions.

Per residue pLDDT scores were obtained from the B-factor column within the PDB file for each predicted structure. We calculated the average pLDDT score of amino acid residues that were five amino acids both upstream and downstream of the mutated residue (as well as the mutated residue itself), as well as those residues that had previously been deemed as falling within a 5 Å radius of the mutated residue during the structural disruption analysis. This process was performed on the wild-type and mutated ColabFold-predicted subunits, before the % change in average pLDDT scores for each local proximity metric from wild-type to mutated nAChR subunit was calculated. This was inspired by Kabir et al. [28]

### 2.5. Measuring Variant-Induced Structure Prediction Quality Changes

pTM scores for individual wild-type and mutated structures were taken from the ColabFold output file, before the % change in pTM was calculated in each case.

### 2.6. AlphaMissense

AlphaMissense variant pathogenicity prediction repositories were downloaded from https://console.cloud.google.com/storage/browser/dm_alphamissense;tab=objects?prefix=&forceOnObjectsSortingFiltering=false (accessed on 17 May 2024). CHRNB1, CHRND, and CHRNE variant predictions were manually obtained from the ‘AlphaMissense_aa_substitutions.tsv.gz’ file. This file did not use the canonical CHRNA1 isoform (UniProt ID: P02708-2), and so CHRNA1 mutations were instead obtained from the ‘AlphaMissense_isoforms_aa_substitutions.tsv.gz’ file.

### 2.7. AlamutVP

We used the Open Gene tab in Alamut Visual Plus v1.7.1 to load the MANE Select gene transcript within the GRCh37 genome for each CHRN gene (RefSeq; CHRNA1: NM_000079.4 CHRNB1: NM_000747.3, CHRND: NM_000751.3, CHRNE: NM_000080.4). For each variant, we queried the variant using corresponding cDNA coordinates. Suggested ACMG classifications were subsequently generated for each variant—ACMG standards were grouped such that ‘Pathogenic’ (score ≥ 10) and ‘Likely Pathogenic’ (score between [6, 9]) were both considered as ‘Pathogenic’ for our analysis. Variants with a predicted pathogenicity class rated as ‘Uncertain Significance’ (score between [0, 5]) were considered ‘Ambiguous’. Variants predicted as either ‘Likely Benign’ (scores between [−6, −1]), and ‘Benign’ (score ≤ −7) were considered ‘Non-pathogenic’ for our analysis.

### 2.8. EVE

EVE variant pathogenicity predictions were obtained manually from https://evemodel.org. Canonical CHRN subunit sequences (accessed on 17 May 2024) (described in Section 2.2) were used for all subunits other than CHRNA1, for which the non-canonical P02708 sequence had to be used—variant nomenclature was adjusted accordingly.

### 2.9. Statistical Analyses

All statistical analyses were performed in GraphPad prism, version 10.1.0. All ColabFold data was checked for normality using the Shapiro–Wilk test; for the majority of datasets that were non-normally distributed, a Mann–Whitney U test was used to compare pathogenic and non-pathogenic variants, whilst the Kruskal–Wallis test was used to compare between individual CHRN CMS subgroups, and non-pathogenic variants. In each case, we report the median and interquartile range of this data. The % change in global pLDDT data was normally-distributed; therefore, Welch’s unpaired *t*-test was performed to compare pathogenic and non-pathogenic variant sets, whilst Welch’s ANOVA was used for multiple group comparisons for this metric. We report the mean and standard deviation for these data. Where significant results were found in Kruskal–Wallis tests, a post hoc Dunn’s test was used to perform pairwise multiple comparisons and reveal group-specific significant differences. Statistical significance was set at 0.05. Receiver Operating Characteristic (ROC) analysis was also performed in GraphPad prism using the Wilson–Brown method [Appendix A: Raw Data, Results Data].

## 3. Results

### 3.1. Variant-Induced ColabFold-Predicted Structural Disruption

#### 3.1.1. Global Structural Disruption

Firstly, we analyzed the ColabFold-predicted global structural disruption caused by a variant of the nAChR, recording the root mean square deviation (RMSD) between ColabFold-predicted wild-type and mutated nAChR subunits. RMSD is the predominant quantitative measure of structural similarity between distinct molecular structures, giving the global structural disruption elicited by a variant (Figure 1A).

We found no significant differences when comparing the RMSD between pathogenic and non-pathogenic variant groups using a Mann–Whitney U test (*p* = 0.3885, Figure 1B), or between RDS, SCS, FCS groups with non-pathogenic variants using the Kruskal–Wallis test (*p* = 0.8345, Figure 1C).

#### 3.1.2. Local Structural Disruption

For a more nuanced measure of predicted structural disruption, we measured the variant-induced structural disruption to the immediate area surrounding the variant site. AlphaFold2 has been demonstrated as sensitive enough to predict the local structural effects of variants, with local distance difference (LDD) scores shown to correlate between experimental and AlphaFold2-predicted pairs of wild-type and mutated proteins [21]. Similarly to RMSD, LDD scores provide a measure of average distance between respective residues in two different structures following alignment, but for atoms/residues within pre-defined localities. We recorded LDD scores between ColabFold-predicted wild-type and mutated nAChR subunit structures by sequence as well as structural proximity—using five amino acids upstream and downstream of the mutated residue as well as the mutated residue itself (10aa LDD), and all amino acids within 5 Å of the mutated residue (5 Å LDD) (Figure 1D).

We performed a Mann–Whitney U test to compare both LDD scores between pathogenic and non-pathogenic variants, finding no statistically significant difference for either metric (10aa *p*-value = 0.8926, 5 Å *p*-value = 0.8804) (Figure 1E,F). We then performed a Kruskal–Wallis test to compare both LDD scores between the different CHRN CMS subgroups and non-pathogenic variants, again finding no statistically significant difference for either LDD score (10aa *p*-value = 0.7738, 5 Å *p*-value = 0.8407) (Figure 1G,H).

### 3.2. Variant-Induced ColabFold Prediction Confidence Change

#### 3.2.1. Global Prediction Confidence Change

As well as providing a protein structure prediction, ColabFold also produces a per-residue confidence metric, the predicted local distance difference test, pLDDT, which reflects its own confidence in the predicted position of individual amino acids. Previously proposed as a potential proxy measure for positional stability [18,29], we also wanted to measure the impact that variants of different CMS classifications had on ColabFold’s global average prediction confidence, thus measuring the % change in average global pLDDT, from the wild-type, to mutated nAChR subunit predictions.

We compared this metric between pathogenic and non-pathogenic variants using Welch’s unpaired *T*-test, finding no statistically significant difference between the two groups (*p*-value = 0.7832) (Figure 2A). Subsequent comparison of this metric between CHRN CMS subgroups and non-pathogenic variants using Welch’s ANOVA also revealed no statistically significant differences (*p*-value = 0.3593) (Figure 2B).

#### 3.2.2. Local Prediction Confidence Change

As with our structural change metrics, we also deemed it sensible to measure not just how a variant affects the confidence of ColabFold in its prediction of the entire protein structure, but also how it alters local structure prediction confidence. Therefore, we also measured the % change in average pLDDT from wild-type to mutated nAChR subunit predictions at two local proximity definitions—for the five amino acids directly upstream and downstream of the mutated residue as well as the mutated residue itself (% change in average 10aa pLDDT), and all amino acids within a 5 Å proximity of the mutated residue (% change in average 5 Å pLDDT) (Figure 2C).

We compared both of these metrics between the combined pathogenic variant group with non-pathogenic variants using a Mann–Whitney U test, finding no statistically significant difference between the groups in either of the two local prediction confidence change measures (10aa *p*-value = 0.7634, 5 Å *p*-value = 0.9569) (Figure 2D,E).

However, when comparing these metrics between each of the RDS, SCS, FCS, and non-pathogenic variant groups using a Kruskal–Wallis test, a statistically significant result was found for the % change in average 10aa and 5 Å pLDDT (10aa: *p* = 0.0083 **, Figure 2F; 5 Å: *p* = 0.0185 *, Figure 2G). A post hoc Dunn’s test subsequently revealed that for both metrics there was a statistically significant difference between the RDS and SCS variant groups (adjusted 10aa *p* = 0.0066 **, adjusted 5 Å *p* = 0.0126 *). In both cases, no other pairwise comparisons were statistically significant.

Following this finding, and considering the relatively low number of FCS variants in our dataset (thus limiting statistical power), we combined the SCS and FCS variants into one pathogenic ‘kinetic’ variant group (i.e., variants that alter the kinetic properties of the receptor as opposed to RDS variants that primarily affect the receptor’s stability). We then performed an additional Kruskal–Wallis test to compare the % change in average 10aa pLDDT and % change in average 5 Å pLDDT, across pathogenic deficiency variants, pathogenic ‘kinetic’ variants, and non-pathogenic variants. In both tests, a greater statistically significant difference was revealed (10aa: *p* = 0.0031 **, Figure 2H; 5 Å *p* = 0.0079 **, Figure 2I). A post hoc Dunn’s test thereafter revealed that in both cases, the statistically significant difference existed between the RDS, and the new ‘kinetic’ variant groups (adjusted 10aa *p* = 0.0021 **, adjusted 5 Å *p* = 0.0056 **). No other statistically significant differences were identified.

### 3.3. Variant-Induced ColabFold Prediction Quality Change

A further output metric that ColabFold provides with each structure prediction is the pTM, which represents the quality of protein structure prediction relative to an existing solved target structure [30] (with this step forming an indispensable step of the ColabFold algorithm). Whilst predominantly a measure of topological accuracy, pTM scores have also been associated with dynamic protein information [31], and therefore we deemed it relevant to assess the effect that variants of the nAChR will have on this ColabFold output value.

We used a Mann–Whitney U test to compare this metric between pathogenic and non-pathogenic groups, with no statistically significant difference between the two groups (*p* = 0.6513) (Figure 3A).

We then compared the pTM data between individual CHRN CMS subgroups, and non-pathogenic variants using a Kruskal–Wallis test. No statistically significant difference was found (*p* = 0.3689) (Figure 3B).

### 3.4. AlphaMissense Variant Pathogenicity Prediction

Following the release of AlphaMissense, we wanted to analyze its acclaimed ability to accurately predict missense variant pathogenicity. In our rigorous dataset of 61 CHRN variants, AlphaMissense predicted the pathogenicity of 39 variants correctly (63.93%), rated 8 variants as ambiguous (13.11%), and predicted 14 variants incorrectly (22.95%, Figure 4A).

Specifically for the 42 pathogenic variants of the nAChR, AlphaMissense correctly predicted 31 as pathogenic (73.81%), 6 as ambiguous (14.29%), and classified 5 variants incorrectly as benign (11.90%, Figure 4B), whereas for the 19 non-pathogenic variants of the nAChR, AlphaMissense correctly predicted 8 of these variants as benign (42.11%), 2 as ambiguous (10.53%), and 9 incorrectly as pathogenic (47.37%, Figure 4C) [Appendix A: AlphaMissense Results].

### 3.5. AlphaMissense Comparison with AlamutVP and EVE

Finally, in order to provide a benchmark with which to compare our AlphaMissense results, we assessed the ability of AlamutVP and EVE to predict the pathogenicity of our nAChR variants. In total, AlamutVP predicted 17 of the variants correctly (27.87%), 43 as ambiguous (70.49%), and 1 variant incorrectly (1.64%, Figure 4A). EVE did not provide a prediction for the CHRNA1 S289I pathogenic variant—however, for the 60 pathogenicity predictions provided, EVE predicted 17 correctly (28.33%), 29 as ambiguous (48.33%), and 14 incorrectly (23.33%, Figure 4A). Regarding pathogenic variants, AlamutVP predicted 17 correctly (40.48%), 25 ambiguously (59.52%), and 0 incorrectly (0.00%), whilst EVE predicted 12 correctly (29.27%), 18 as ambiguously (43.90%), and 11 incorrectly (26.83%) (Figure 4B). For non-pathogenic variants, AlamutVP predicted 0 correctly (0.00%), 18 ambiguously (94.74%) and 1 incorrectly (5.26%), and EVE predicted 5 correctly (26.32%), 11 ambiguously (57.89%), and 3 incorrectly (15.79%) (Figure 4C). ROC analysis can be used to directly evaluate and compare the accuracy of different diagnostic tests; thus, we finally performed ROC analysis to compare AlphaMissense, AlamutVP, and EVE. Despite AlphaMissense predicting a greater proportion of our variant dataset correctly, AlamutVP had the highest discriminatory power for our dataset based on ROC analysis, with an area under the curve (AUC) of 0.7688, compared to AlphaMissense (0.7231), and EVE (0.5944) (Figure 4D–F).

Analysing variant effects on the ColabFold CHRN structure prediction, prediction confidence, and prediction quality does not yield any pathogenicity indicative metric. However, AlphaMissense predicts variant pathogenicity with 63.93% accuracy on our dataset—a much greater proportion than AlamutVP (27.87%) and EVE (28.33%) [Appendix A: AlphaMissense vs. Alamut and EVE Results].

## 4. Discussion

Currently, the determination of novel CMS genetic variant pathogenicity relies on manual diagnostic techniques that can only be performed by a handful of diagnostic centers worldwide. Identification of new, reliable, and accessible CMS diagnostic tools would permit faster treatment optimization for patients globally. Following AlphaFold2’s release in 2021, and the subsequent development of derivative programs such as ColabFold, ambiguity has surrounded the application of these protein structure prediction software programs to functional variant studies. More recently, AlphaMissense was reported to demonstrate great accuracy in this challenge of protein variant pathogenicity classification on a broad ClinVar dataset. However, no studies to our knowledge have used clinically and experimentally validated data to test whether ColabFold and AlphaMissense are useful tools for predicting missense variant pathogenicity. In this study, we compared the accuracy of AlphaMissense with the established pathogenicity prediction programs AlamutVP and EVE.

As AlphaFold2/ColabFold predicts structures based on peptide sequence homology to published experimental structures, it was unlikely that a single missense variant would significantly disrupt the predicted overall structure. Therefore, we investigated whether these algorithms could predict significant structural disruption in the vicinity of the variant. Indeed, there were no statistical significances between any of the data outputs for the overall structure (RMSD, pLDDT, pTM), but we did observe some statistically significant differences between the local (10aa and 5 Å) pLDDT scores, with RDS pathogenic variants producing significantly higher values than SCS or the combined ‘kinetic variants’ group. This is intriguing, because RDS variants would be expected to disrupt structural stability, and pLDDT has been thought to serve as a proxy for residue stability [18,29,32]. Conversely, kinetic variants would be expected to stabilize a particular conformation of the protein—either an active state, or an inactive state. However, there was no significant difference between the RDS variants and non-pathogenic variants, meaning that this metric was not useful for determining RDS variant pathogenicity. Investigations into local structural changes (10aa and 5 Å LDD scores) did not show any significant differences between groups.

We next showed that the more recently released AlphaMissense model is able to predict CHRN variant pathogenicity accurately 63.93% of the time. Our results replicate similar findings to other validatory studies of AlphaMissense [33], in that we note AlphaMissense’s propensity to overestimate variant pathogenicity. In our dataset, AlphaMissense correctly predicted 73.81% of pathogenic variants as pathogenic, and only 11.90% as non-pathogenic. Conversely, AlphaMissense more commonly predicted non-pathogenic variants of the nAChR as pathogenic (47.37%) than as non-pathogenic (42.11%).

Crucially, we compared our AlphaMissense results with the ability of AlamutVP and EVE to predict variant pathogenicity using our dataset. AlphaMissense does predict a significantly greater proportion of variants correctly (63.93%) compared to Alamut (27.87%), and EVE (28.33%). However, ROC analysis demonstrated a greater AUC for AlamutVP (0.7688) compared to AlphaMissense (0.7231) and EVE (0.5944); this discrepancy could perhaps reflect overly conservative classification parameters within AlamutVP and the ACMG classification used, suggesting that a lower threshold for pathogenic classification prediction within AlamutVP could be employed to improve prediction accuracy. Considering the % of correct pathogenicity predictions was our primary measure of accuracy, we deem AlphaMissense in its current state a more useful tool than both AlamutVP and EVE. Additionally, the ability of AlphaMissense to extend into novel missense variants, as opposed to, for example, EVE (which is only available for a certain defined set of genetic variants), further strengthens this potential use case. Other tools, including DEOGEN2 [34], FATHMM [35], and CADD [36], are alternative variant pathogenicity prediction programs that could be utilized. Such algorithms, however, have until very recently relied on traditional machine learning and statistical approaches, and thus fail to utilize the potential of advanced deep learning models, which are central to AlphaMissense, and which are inherently more able to capture representations underlying the complexity of protein structures and pathogenicity predictions for novel variants.

AlphaMissense thus constitutes a welcome extension to prior structural prediction software, targeted to this specific problem of variant pathogenicity prediction. However, whilst initial genome-wide results are encouraging, our results provide reason for caution regarding the clinical utility of AlphaMissense. How to improve such programs remains a somewhat empirical question—certainly, the relative scarcity of training data on which to train such programs remains a challenge upon which attention should be focused. Indeed, the very recent use of a more expansive training dataset in the development of BaseFold has been shown to dramatically improve protein structure prediction accuracy from AlphaFold2 levels [37], and the integration of improved ligand docking, as well as protein–protein interaction information in Alphafold3 could yet improve pathogenicity prediction. The incorporation of such increasingly accurate protein structural information within variant pathogenicity predictions, as with AlphaFold2 structures and AlphaMissense, is likely to improve the accuracy of such programs, potentially to clinically useful levels.

We believe our study to be one of the first investigations using a clinically validated dataset into the ability of new-age protein structure prediction software, as well as the recently released AlphaMissense, to predict functional variant effects. By comparing the ColabFold signature of variants across different CHRN CMS subgroups and non-pathogenic variants, we demonstrate that for all the metrics tested, substantial variance within and overlap between variant groups means that we fail to provide evidence for the incorporation of ColabFold into CMS diagnostic procedures. We also show that whilst the recently released AlphaMissense does predict a greater proportion of variants accurately than AlamutVP and EVE, it fails to reach the degree of accuracy reported in seminal studies using large datasets of ClinVar variants [23]. The muscle nAChR may be a particularly challenging protein for prediction algorithms, as it is a heteromultimeric membrane protein with numerous conformational states. However, structural prediction algorithms are developing rapidly, and with the recent release of AlphaFold3, the ability to look at predicted differences in protein complex formation, conformational stability, and ligand docking efficiencies are all on the horizon. Therefore, although we show here that ColabFold and AlphaMissense were unable to predict CMS missense variant pathogenicity with a clinically useful level of accuracy, structure-based in silico approaches hold promise for helping diagnosis of CMSs and other genetic diseases more broadly.

## 5. Conclusions

Our results provide important insights into the evolving role of AI and in silico prediction tools in genetic variant pathogenicity, specifically within the context of CMS. Our evaluation of ColabFold and AlphaMissense highlights both the potential and limitations of such computational models. Whilst ColabFold’s protein structure predictions failed differentiate between pathogenic and non-pathogenic variants in any of the parameters we investigated, AlphaMissense demonstrated a comparatively higher accuracy (63.93%) in predicting variant pathogenicity, surpassing the existing tools AlamutVP and EVE. However, AlphaMissense showed a high false positive rate, predicting nearly half of the non-pathogenic variants as pathogenic. Therefore, we conclude that it is not currently reliable enough for clinical use.

## Figures and Tables

**Figure 1 biomedicines-12-02549-f001:**
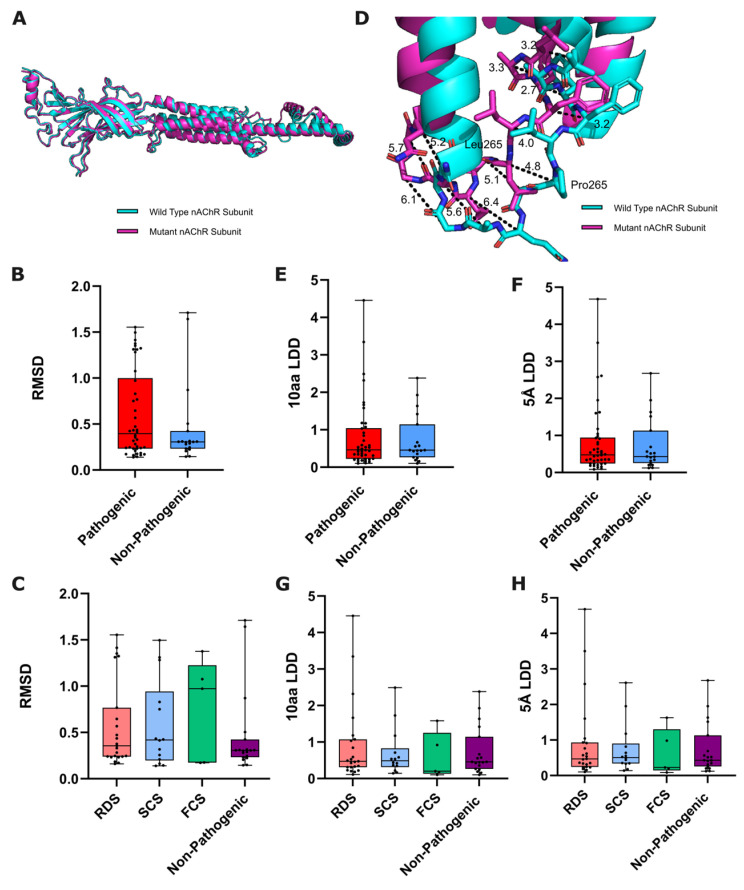
Structural-based metrics of nAChR subunit variant predicted effect. (**A**) Alignment of predicted wild-type and mutated nAChR subunits, from which RMSD is calculated. Both wild-type and mutated CHRN subunits were loaded into PyMol, before the ‘align’ feature was used, consequently providing an RMSD score between the two structures. (**B**) Box plots comparing the RMSD of pathogenic and non-pathogenic variants from WT (*p* = 0.3885: Mann–Whitney U test, n = 42/19), and (**C**) between functional CHRN CMS subgroups and non-pathogenic variants (*p* = 0.8345: Kruskal–Wallis test, n = 23/14/5/19). (**D**) Example measurements of a variant’s local structural effect on corresponding residues in wild-type and P265L mutated CHRNE (from which LDD scores are derived). Box plots showing 10aa/5 Å LDD scores of ColabFold-predicted wild-type and mutated nAChR subunits, compared between (**E**,**F**) pathogenic and non-pathogenic variants (10aa *p* = 0.8926, 5 Å *p* = 0.8804: Mann–Whitney U test, n = 42/19), and (**G**,**H**) between functional CHRN CMS subgroups and non-pathogenic variants (10aa *p* = 0.7738, 5 Å *p* = 0.8407: Kruskal–Wallis test, n = 23/14/5/19).

**Figure 2 biomedicines-12-02549-f002:**
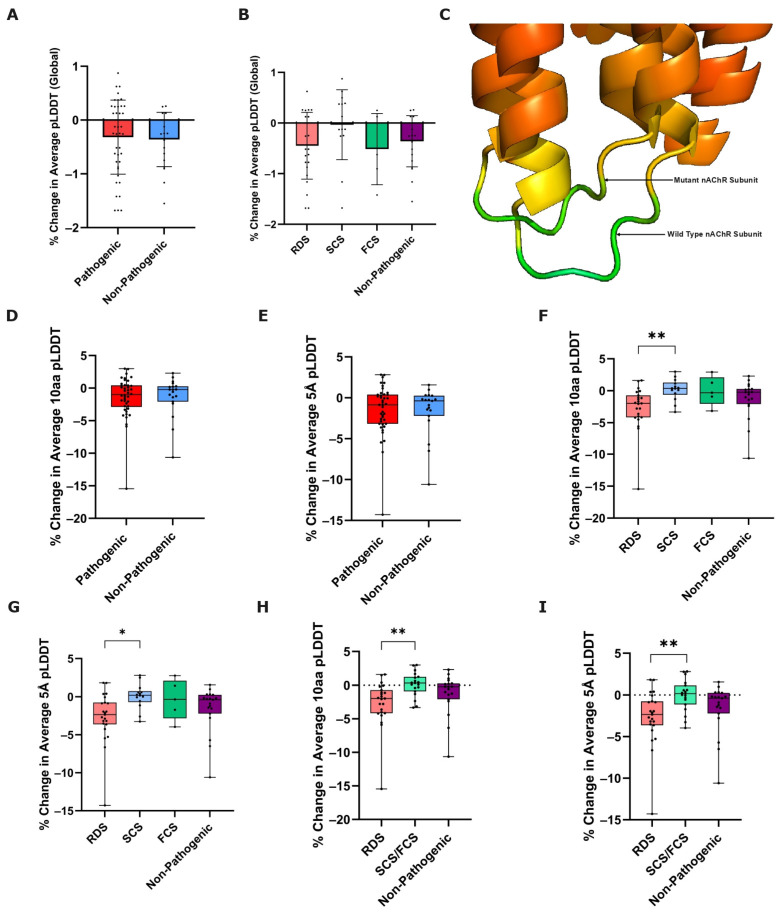
Prediction confidence-based metrics of nAChR subunit variant effect. % change in global pLDDT was calculated from ColabFold-predicted wild-type to mutated nAChR subunit structures, and compared between (**A**) pathogenic and non-pathogenic variants (*p* = 0.7832: Welch’s unpaired *T*-test, n = 42/19), and (**B**) functional CHRN CMS subgroups and non-pathogenic variants (*p* = 0.3593: Welch’s ANOVA, n = 23/14/5/19). Error bars represent mean ± SD. (**C**) Representation of differences in pLDDT between WT and P265L mutant CHRNE, rainbow-colored based on B-factor (confidence) values from ColabFold .pdb file output, red = more confident to green = less confident. Box plots showing % change in 10aa and 5 Å pLDDT from ColabFold-predicted wild-type to mutated nAChR subunit structures, compared between pathogenic and non-pathogenic variants (10aa *p* = 0.7634, 5 Å *p* = 0.9569: Mann–Whitney U test, n = 42/19) (**D**,**E**), and between functional CHRN CMS subgroups and non-pathogenic variants (10aa *p* = 0.0083 **, 5 Å *p* = 0.0185 *, Kruskal–Wallis test, n = 23/14/5/19) (**F**,**G**). Post hoc Dunn’s Test for the % change in average 10aa/5 Å pLDDT revealed a statistically significant difference between RDS and SCS variants for both metrics (10aa adjusted *p*-value = 0.0066 **, 5 Å adjusted *p*-value = 0.0126 *). To increase statistical power of comparisons, SCS and FCS variants were subsequently combined into a ‘kinetic’ variant group. Box plots showing % change in average (**H**) 10aa pLDDT, and (**I**) 5 Å pLDDT, compared between RDS, kinetic (SCS+FCS), and non-pathogenic variants (10aa *p* = 0.0031 **, 5 Å *p* = 0.0079 **: Kruskal–Wallis test, n = 23/19/19). Post hoc Dunn’s Test for the % change in both 10aa/5 Å average pLDDT revealed a statistically significant difference between RDS and kinetic (SCS+FCS) variants (10aa adjusted *p*-value = 0.0021 **, 5 Å adjusted *p*-value = 0.0056 **).

**Figure 3 biomedicines-12-02549-f003:**
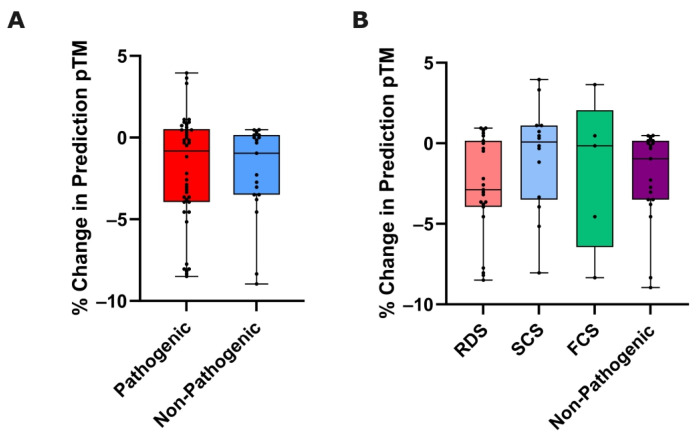
% Change in pTM as a metric of nAChR subunit variant effect. Box plots showing % change in pTM from wild-type to mutated ColabFold-predicted nAChR subunit, compared between (**A**) pathogenic and non-pathogenic variants (*p* = 0.6513: Mann–Whitney U test, n = 42/19), and (**B**) functional CHRN CMS subgroups and non-pathogenic variants (*p* = 0.3689: Kruskal–Wallis test, n = 23/14/5/19).

**Figure 4 biomedicines-12-02549-f004:**
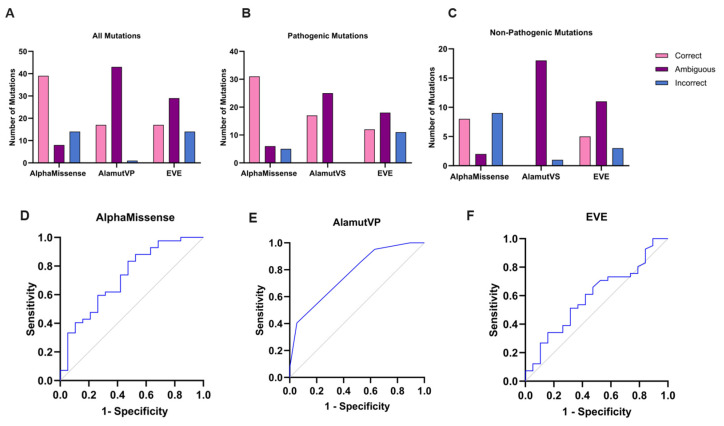
AlphaMissense comparison with AlamutVP and EVE. Column charts comparing the number of variants that AlphaMissense, AlamutVP, and EVE predicted as correct/ambiguous/incorrect, for (**A**) all variants combined, (**B**) for pathogenic variants only, and (**C**) for non-pathogenic variants only. ROC analyses demonstrated that AlamutVP had the greatest area under the curve (AUC); 0.7688 (**D**), compared to AlphaMissense (0.7231) (**E**), and EVE (0.5944) (**F**).

**Table 1 biomedicines-12-02549-t001:** ColabFold parameter values utilized in our study. From the ColabFold output, the model with the highest global pLDDT score was taken as the ColabFold prediction for that given sequence.

Parameter	Value
MSA Mode	MMseqs2 (UniRef + Environmental)
Number of Models	5
Number of Recycles	3
Stop at Score	100
Use Amber	No
Use Templates	No

## Data Availability

The human variant data included in this study will be deposited in ClinVar prior to publication. The data collected and analyzed during this study can be obtained from the corresponding author upon request.

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
