# Peer review of "Assessing the Utility of ColabFold and AlphaMissense in Determining Missense Variant Pathogenicity for Congenital Myasthenic Syndromes"

_biomedicines, 2024, doi:10.3390/biomedicines12112549_

Round 1
Reviewer 1 Report
Comments and Suggestions for Authors
Summary:
I appreciate the aim of your work and agree with the concept that in silico tools can provide a comprehensive understanding of the molecular mechanisms involved in protein folding. The methodological aspects are well-detailed; however, the Introduction requires additional information (specified below). Additionally, the manuscript formatting and design must adhere to the MDPI’s Instructions for Authors.
Introduction:
This section needs to be formatted in line with the MDPI’s Instructions for Authors. Moreover, when citing MIM phenotype entries, ensure that the specific phenotype linked to the entry code is indicated, rather than the OMIM entry associated with the gene. Addressing this will improve the accuracy of your work.
- Incorporate references discussing the recent developments related to AlphaFold3.
- Additionally, consider mentioning alternative software that can be used for protein modeling to provide a broader perspective on the available tools.
Materials and Methods:
- The access dates to the various online databases (e.g., UniProt, ClinVar) should be mentioned, as they are necessary for reproducibility.
- The access date for ColabFold must also be specified.
- Line 110: The phrase "our CMS referral service" is unclear. Specify the institution or dataset provider, as well as the source referred to on Line 87 of the Introduction.
- Line 111: I believe you meant to use the term “ensuring” instead of “ensuing.”
- Indicate the version of PyMOL used in the study. The explicit mention of the commands is appreciated and adds clarity to the methods section.
Subsection 2.6:
- How did you carry out the pathogenicity prediction using the following link: https://console.cloud.google.com/storage/browser/dm_alphamissense;tab=objects?prefix=&forceOnObjectsSortingFiltering=false? Did you download the entire repository?
- Have you compared the results obtained from AlphaMissense with those reported by ProtVar (https://www.ebi.ac.uk/ProtVar/)? This tool is user-friendly and based on AlphaMissense predictions.
Results:
- Table 1: The description needs additional detail. For instance, indicate the “best model” selected from the ColabFold output.
- Figure 1: Provide more information on how you performed the alignment, particularly specifying the use of PyMOL software.
Discussion:
Could the fact that some scores (e.g., EVE) are available only for known genetic variants be a limitation for future studies? Your study's objective is to compare predictions made by AlphaMissense with those for known variants. However, for a novel missense variant, would the absence of accurate pathogenicity predictions present a limitation?
- Have you considered using other well-known tools for predicting missense mutations, such as DEOGEN2, FATHMM, or CADD?
- The references need to be formatted in accordance with MDPI’s Instructions for Authors.
Line 410:
I agree with your statement.
Conclusion
Although a conclusion is not mandatory, I highly recommend including one. This would strengthen your results and position them within the modern landscape of artificial intelligence and in silico predictors, highlighting their affordability and utility in future studies.
Comments on the Quality of English LanguageI appreciate the aim of your work and agree with the concept that in silico tools can provide a comprehensive understanding of the molecular mechanisms involved in protein folding. The methodological aspects are well-detailed; however, the Introduction requires additional information (specified below). Additionally, the manuscript formatting and design must adhere to the MDPI’s Instructions for Authors.
Author Response
Responses to reviewer
We would like to thank the reviewer for their detailed and considered comments on our manuscript. Here are our responses to their specific comments (comments are italicized):
Introduction:
This section needs to be formatted in line with the MDPI’s Instructions for Authors.
We have now added a paragraph at the end of the introduction that states the hypotheses, significance of work, and main conclusions.
Moreover, when citing MIM phenotype entries, ensure that the specific phenotype linked to the entry code is indicated, rather than the OMIM entry associated with the gene. Addressing this will improve the accuracy of your work.
MIM codes have now been added.
- Incorporate references discussing the recent developments related to AlphaFold3.
- Additionally, consider mentioning alternative software that can be used for protein modeling to provide a broader perspective on the available tools.
Following the introduction of AlphaFold2, we have added a section that highlights alternative protein modelling tools OmegaFold/ESMFold, as well as RosettaFold/AlphaFold3 that permit modelling of protein-ligand interactions.
Materials and Methods:
- The access dates to the various online databases (e.g., UniProt, ClinVar) should be mentioned, as they are necessary for reproducibility.
- The access date for ColabFold must also be specified.
Access dates have now been added to the manuscript for UniProt and ColabFold (4th September 2022)
- Line 110: The phrase "our CMS referral service" is unclear. Specify the institution or dataset provider, as well as the source referred to on Line 87 of the Introduction.
We apologise for this oversight, and have clarified in both the introduction and methods that the dataset is from the UK national congenital myasthenia referral service.
- Line 111: I believe you meant to use the term “ensuring” instead of “ensuing.”
To clarify, we do mean to use the term “ensuing” in this sentence.
- Indicate the version of PyMOL used in the study. The explicit mention of the commands is appreciated and adds clarity to the methods section.
We have stated the version of PyMOL used in the methods section, and the command used in both methods, and figure 1
Subsection 2.6:
- How did you carry out the pathogenicity prediction using the following link: https://console.cloud.google.com/storage/browser/dm_alphamissense;tab=objects?prefix=&forceOnObjectsSortingFiltering=false? Did you download the entire repository?
We have added extra details here to clarify – we downloaded the entire repositories for the genes of interest in two files.
- Have you compared the results obtained from AlphaMissense with those reported by ProtVar (https://www.ebi.ac.uk/ProtVar/)? This tool is user-friendly and based on AlphaMissense predictions.
We compared the mutations in our database with ProtVar and obtained the same results as what we downloaded originally from Alphamissense for all genes except CHRNA1. ProtVar uses the non-canonical P02708-1 isoform (the P3A isoform, known to have aberrant function) for CHRNA1, and outputted 'UniProt accession not found P02708-2' when we tried to use the functional P02708-2 isoform. To obtain AlphaMissense predictions for the P02708-2, we had to download the separate isoform files from AlphaMissense. This has been explained in the revised methods section.
Results:
- Table 1: The description needs additional detail. For instance, indicate the “best model” selected from the ColabFold output.
Have added this to the table 1 description.
- Figure 1: Provide more information on how you performed the alignment, particularly specifying the use of PyMOL software.
We have added more details to the figure 1 legend, specifying the use of the 'align' feature in PyMol.
Discussion:
Could the fact that some scores (e.g., EVE) are available only for known genetic variants be a limitation for future studies? Your study's objective is to compare predictions made by AlphaMissense with those for known variants. However, for a novel missense variant, would the absence of accurate pathogenicity predictions present a limitation?
We agree with this reviewer that it is a limitation of EVE to only have predictions for known variants, and have added this consideration to the revised discussion.
- Have you considered using other well-known tools for predicting missense mutations, such as DEOGEN2, FATHMM, or CADD?
We have added mentions of these tools in the revised discussion.
- The references need to be formatted in accordance with MDPI’s Instructions for Authors.
We have received assurances from the editors that if this manuscript is accepted, the editorial office will re-layout the full text and references for us.
Line 410: I agree with your statement.
We are glad that this reviewer agrees with this statement
Conclusion
Although a conclusion is not mandatory, I highly recommend including one. This would strengthen your results and position them within the modern landscape of artificial intelligence and in silico predictors, highlighting their affordability and utility in future studies.
We’d like to thank the reviewer for this advice, and have added a conclusions section to the revised manuscript.
Reviewer 2 Report
Comments and Suggestions for Authors
In this manuscript, Finlay et al. assess the ColabFold and AlphaMissense tools as diagnostic aids for congenital Myasthenic Syndromes (CMS) using a collection of 61 clinically validated CHRN variants. Overall, the manuscript is well-written, clearly outlining the methods, experiments, conclusions, and discussions. The data visualization is also commendable. I have a few minor comments to consider before full acceptance for publication in the Biomedicines journal:
- In the introduction section (line 88), the three subgroups of CHRN variants are presented using abbreviations without prior explanation. Their meanings are unclear, which complicates the understanding of the discussion between lines 274 and 276, where two of the subgroups are combined.
- In the methods section (lines 150 to 153), please include a citation if applicable. Additionally, is there a specific reason for using "5" for amino acids and "5A" for radius? Is this a common convention in similar analyses?
- There is a typo on line 240 that needs correction.
Author Response
Responses to reviewer:
We’d like to thank the reviewer for their positive and considered comments to our manuscript. Here are our responses to their specific comments (comments are italicized):
- In the introduction section (line 88), the three subgroups of CHRN variants are presented using abbreviations without prior explanation. Their meanings are unclear, which complicates the understanding of the discussion between lines 274 and 276, where two of the subgroups are combined.
We apologise for this oversight, and have explained the abbreviations in the introduction of revised manuscript.
- In the methods section (lines 150 to 153), please include a citation if applicable. Additionally, is there a specific reason for using "5" for amino acids and "5A" for radius? Is this a common convention in similar analyses?
Looking at 5 amino acids up and downstream of the site of mutation as well as 5A radius was used in a previous study, and we thought that it was a good way of looking at local differences. We have acknowledged the previous study, and added the reference accordingly.
- There is a typo on line 240 that needs correction.
We would like to thank the reviewer for pointing this out, and have corrected this typo in the revised manuscript
Round 2
Reviewer 1 Report
Comments and Suggestions for Authors
The manuscript is ready for publication, but it will need to be formatted to align with MDPI’s specific template requirements.